# COMPETITIVE-COLLABORATIVE GAN WITH PERFORMANCE GUARANTEE

## ABSTRACT

Generative Adversarial Networks (GANs) generate data based on a competition game to minimize the distribution distance between existing and new data. However, such a competition game falls short when insights about data distributions beyond their authenticity are imperative, such as in multi-modal generation and image super resolution. In recognition of the limitations inherent to the pure-competition mechanism, we introduce CCGAN, a Collaborative-Competitive Generative Adversarial Network scheme to enable data generation with additional knowledge beyond the provided dataset distribution. For theoretically preserving the equilibrium point and numerically avoiding training collapse issue, we show the need to convert the regularization term into a divergence metric, so that the modified GAN is well-defined in game theory. By harmonizing the competition and collaboration losses in CCGAN, we effectively reduce the degree complexity of solving the optima, facilitating the establishment of a closed-form equilibrium point. This equilibrium point serves as a guidance for training and hyper-parameter tuning, resulting in consistently high-quality generations. Meanwhile, the regularization breaks the mutual dependency between the generator and discriminator. This newfound independence empowers the CCGAN to explore a broader parameter space, effectively mitigating the training collapse issue. To validate the capabilities of CCGAN, we design comprehensive experiments across four publicly available datasets and systematically compare CCGAN against a range of baseline models. The experiments demonstrate the efficacy of CCGAN on generating satisfactory samples tailored to specific requirements, particularly when applied to the generation of images featuring regularly shaped objects.

## 1 INTRODUCTION

Generative Adversarial Networks (GANs) Goodfellow et al. (2014; 2020) empower the creation of deceptively real data, showing success across diverse data types, such as images Heusel et al. (2017), text Haidar & Rezagholizadeh (2019), and speech Eskimez et al. (2020). In GAN's framework, the adversarial dynamic between the generator and discriminator gives rise to a pure-competition game scenario (see Figure 1 (a)): the generator iteratively refines its ability to create samples that are indistinguishable from real data. It does so by learning to deceive the discriminator, who also iteratively refines its ability to distinguish between real (authentic) and fake (synthetic) data.

While GANs gain success in their competitive design to generate synthetic data that follow authentic distribution, this pure-competition nature becomes less effective in certain applications Huang et al. (2022). In such applications, the need extends beyond capturing the authentic data distribution (see Figure 1 (b)). For instance, in multi-modal generation Liu et al. (2021), the focus is on learning one or multiple modes within the data distribution. In image super-resolution Zhu et al. (2020a), one must not only capture the data distribution but also faithfully reproduce the original image. The additional requisites for generation pose challenges for pure-competition GANs, as they lack a inherent incentive for the generator and discriminator to collectively fulfill the additional requisites.

To overcome the challenges inherent in pure-competition GANs, researchers have ventured into the realm of collaboration-competition design of GANs. By re-designing the loss function, one can foster collaboration between the generator and discriminator to collectively fulfill additional requirements regarding generated data Durgadevi et al. (2021). These strategies to reshape the competitive nature of GANs include (1) replacing the standard binary cross-entropy loss with alternative loss functions, such as the least square loss Mao et al. (2017) and Wasserstein loss Arjovsky et al. (2017), (2) introducing regularization terms to promote collaboration between the generator and discriminator (see Figure 1 (c)), such as coordinate GAN Lin et al. (2019) and collaborative GAN (CollaGAN) Lee et al. (2019), and (3) designing multiple generators Zhang et al. (2020) and multiple discriminators Choi & Han (2022) to promote collaboration.

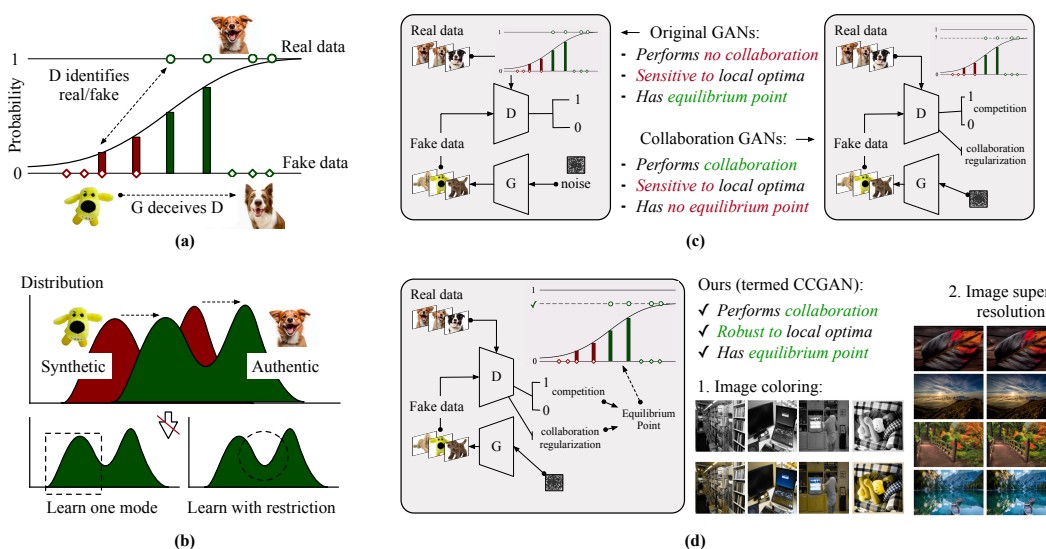

Figure 1: **(a)** Discriminator $D$ learns a binary classification between fake and real data. Generator $G$ learns to produce deceptively real data to fool $D$. **(b)** GAN's adversarial nature is limited in applications requiring knowledge beyond the "authentic" data distribution. **(c)** The architecture of original GAN and GAN variants that conduct a collaboration-competition game. **(d)** The architecture of our proposed model CCGAN and its preliminary experimental results.

However, these collaboration-competition studies have limitations, possibly yielding sub-optimal results. First, they lack a theoretical guarantee. Specifically, the inconsistent integration of the original GAN loss and the collaboration regularization disrupts the original GAN's theoretical guarantee, resulting in a blind model without proper guidance Lin et al. (2019). Second, they can be numerically stuck in local-optima if training collapses Bang & Shim (2021). Specifically, when generator and discriminator attain a state of local optima, achieving further enhancements becomes increasingly arduous due to their mutual dependent relationship. Overall, the absence of a theoretical guarantee and the training collapse issue can result in misguided collaboration and sub-optimal generated data.

To address the aforementioned challenges, which involve three key aspects: (1) pushing the boundary of GAN from pure-competition to collaboration-competition, (2) theoretically deriving the equilibrium point of this collaboration-competition game, and (3) numerically avoiding training collapse, this paper proposes a novel design of regularization for the collaboration need. Specifically, since the existing methods fail to obtain a closed-form equilibrium point due to inconsistent integration between the collaboration and competition loss components, we aim to establish a smooth integration (see Figure 1 (d)), where the collaboration loss function can be transformed into a divergence metric. With this in mind, we present the innovative solution named **C**ollaboration **Co**mpetition-based **G**enerative **A**dversarial **N**etwork (CCGAN). Our main contributions are:

- Employing collaboration-competition mechanism, we establish a unified optimization procedure that can generate high-fidelity samples catering to specific requisites (see Figure 1 (d)). The numerical experiments also reveal one interesting phenomenon:

    - The generated image samples exhibit superior quality when the objects within the images are of regular shape. This advantage can potentially be attributed to the presence of the closed-form equilibrium point.

- We leveraged Jensen–Shannon divergence to develop a smooth integration between the collaboration and competition loss components. It enables the attainment of a global optimality, i.e., the equilibrium point.
- We prove that the generator and discriminator is not mutually dependent in CCGAN. This non-mutual-dependence empowers CCGAN to explore more extensive parameter space, effectively mitigating the issue of training collapse.

Overall, these findings provide both theoretical and empirical insights into collaboration-competition GANs, paving the way for stable and consistently high-quality data generation with specific requirements that extend beyond the provided data distribution.

## 2 RELATED WORK

Related work section considers three lines of intellectual inquiry: (1) recent developments of GAN, (2) GANs with regularization, and (3) collaboration-competition deisgn in machine learning.

**Recent developments of GAN.** Addressing critical challenges identified in GAN Chen (2021), current research trends include the follows. First, there's a concerted effort to enhance training techniques Zhong et al. (2020) and achieve training stability Neyshabur et al. (2017). Second, researchers are working towards the development of interpretable and controllable GANs while ensuring performance guarantees Tripathy et al. (2020). Scalability is another focus, with endeavors to establish large-scale and high-resolution GANs Brock et al. (2018). Moreover, GANs are expanding their horizons beyond image generation to include data generation in diverse domains Chen et al. (2018), the generation of multi-modal data Liu et al. (2021), and cross-domain data translation Reed et al. (2016). Our paper investigates the interconnected landscape of several of these trends, with the overarching goal of advancing GAN research in training, interpretability, and application.

**GANs with regularization.** Building upon the original GAN, researchers have introduced various regularization techniques for a multitude of practical objectives, such as fulfilling additional requirements for generated data Lin et al. (2019); Lee et al. (2019), addressing mode collapse issue Nagarajan & Kolter (2017); Bang & Shim (2021), ensuring training stability Brock et al. (2018), and preserving privacy Wu et al. (2019); Chen et al. (2021). However, the introduction of regularization terms may inadvertently alter the optimization landscape of GAN training, compromising the theoretical guarantee of original GAN Nagarajan & Kolter (2017). Hence, there is a demand for the development of a novel regularization term that simultaneously addresses practical objectives while preserving theoretical guarantees.

**Collaboration-competition design in machine learning.** In machine learning, the notion of collaboration-competition emerges in social network studies Wasserman & Faust (1994); Fu et al. (2008); Lee et al. (2014). The goal was to balance collective advancement and individual progress Lee et al. (2012), promoting cooperation for communal objectives while motivating individuals to excel and outperform their peers Aaldering et al. (2019). Following its success in social networks, the collaboration-competition framework extended its influence to neural network studies Shi et al. (2020); Kopacz et al. (2023). This concept entails coordinating the contributions of various models/networks to collectively boost performance, while preserving a competitive element to drive individual model enhancements. This approach fosters knowledge sharing Gong et al. (2020), information exchange Tan et al. (2019), and collaborative learning among neural networks, resulting in improved generalization, model diversity, and overall network efficacy. In this paper, we leverage the collaboration-competition idea in GAN to combine individual network strengths Poirot et al. (2019) and harness collective intelligence for superior outcomes.

## 3 PROPOSED MODEL

**Pure-competition GANs and challenges.** To present the motivation of our method, we start from analysing the mechanism and challenges in original GAN. The original GAN model Goodfellow et al. (2014) conducts an adversarial training between the generator $G$ and discriminator $D$ as

$$\min_G \max_D V_{\text{GAN}}(D, G) = \mathbb{E}_{\mathbf{x} \sim p_{\text{data}}(\mathbf{x})}[\log(D(\mathbf{x}))] + \mathbb{E}_{\mathbf{z} \sim p_{\mathbf{z}}(\mathbf{z})}[\log(1 - D(G(\mathbf{z})))]. \quad (1)$$

It aims to capture $p_{\text{data}}$, which represents the underlying data distribution of given samples $\mathbf{x}$. For this purpose, $G$ generates fake samples $G(\mathbf{z})$ from random noises $\mathbf{z}$[1], while $D$ acts as a binary classifier to distinguish whether a sample belongs to the given training data or is generated by $G$ (see Figure 1 (a)). From the pure-competition nature of the generator-discriminator relationship, GANs can generate fake data that follow "real" distribution, thus showing success in generating diverse data types. However, there also remains challenges associated with the pure-competition design where learning the "real" distribution is not the only focus Mao et al. (2017); Huang et al. (2022). In particular, certain applications may necessitate the acquisition of knowledge about a single mode within the distribution or a restricted portion of the distribution (see Figure 1 (b)). For example, in the task of multi-modal generation Liu et al. (2021), it requires to learn one or several modes within the data distribution. In image super-resolution Zhu et al. (2020a), it is essential not only to grasp the underlying data distribution but also to replicate the exact original image. For these applications,

---

[1]$\mathbf{z}$ follows a given distribution (e.g., Gaussian) denoted as $p_{\mathbf{z}}(\mathbf{z})$

the focus is understanding distributions beyond its authenticity, thus necessitating another incentive beyond the pure-competition loss function in GAN.

**Collaboration-competition GANs and challenges.** To fulfill the specific need beyond the authenticity of the data distribution, recent studies Lin et al. (2019); Zhang et al. (2020); Lee et al. (2019) have introduced regularization terms to GAN. They are designed to align the objectives of both the generator and discriminator, guiding them towards a common need. The general form reads as

$$\max_D \ V_{\text{GAN}}(D, G) - L_{\text{regularization}}(D, G; \lambda); \ \ \min_G \ V_{\text{GAN}}(D, G) + L_{\text{regularization}}(D, G; \lambda), \quad (2)$$

where $V_{\text{GAN}}(D, G)$ is the original GAN's loss function in Equation (1), $L_{\text{regularization}}(D, G; \lambda)^2$ the loss tailored to address specific needs, and $\lambda$ the hyperparameter.

However, the collaboration-competition game in Equation (2) represents a rough combination of regularization $L_{\text{regularization}}(D, G; \lambda)$ to the original GAN loss $V_{\text{GAN}}(D, G)$. As shown in Theorem 1, this fragmented design compromises the equilibrium point of original GAN. Without the equilibrium point, the model will suffer from misguided collaboration and can yield sub-optimal outcomes. For example, the hyper-parameter ($\lambda$) tuning can become a challenging task devoid of clear guidance or guarantees Kurach et al. (2018), and the haphazard mixing of loss functions can introduce conflicting objectives, hindering convergence and generating subpar outputs Tran et al. (2018).

**Theorem 1.** *[No equilibrium point]. Equation (2) lacks a closed-form equilibrium point when $L_{regularization}(D, G; \lambda)$ is squared loss, and the true data distribution is Gaussian.*

The proof of Theorem 1 can be found in Appendix A. It demonstrates that because of the inconsistent combination of $L_{\text{regularization}}(D, G; \lambda)$ and $V_{\text{GAN}}(D, G)$, the equilibrium equation is a polynomial equation with a degree exceeding five, thus lacking an analytical solution Ramond (2022). To resolve this, it's crucial to "harmonize" $L_{\text{regularization}}(D, G; \lambda)$ and $V_{\text{GAN}}(D, G)$ in Equation (2), ensuring that $L_{\text{regularization}}(D, G; \lambda)$ doesn't introduce extra complexity in finding an equilibrium point.

### 3.1 CCGAN FRAMEWORK

This section presents CCGAN, a GAN model that jointly considers the collaboration requirements and the equilibrium point, aiming to enhance the collaboration, interpretability, and overall optimality of the GAN model. The key design, as depicted in Equation (3)-(5), is chosen to enable the conversion from the loss function to a divergence metric. This transformation can prevent the introduction of extra degrees of complexity when seeking an equilibrium point.

Motivated by the work Mao et al. (2017) and Nowozin et al. (2016), we consider the regularization in Equation (3), which mimics the form of loss function in Equation (1). The corresponding optimization problem between discriminator $D$ and generator $G$ is shown in Equation (4)-(5):

$$L_{\text{regularization}}(D, G; \lambda) = \mathbb{E}_{\mathbf{x} \sim p_{\text{data}}(\mathbf{x})} \log(D(\mathbf{x}) - \lambda) + \mathbb{E}_{\mathbf{z} \sim p_{\mathbf{z}}(\mathbf{z})} \log(D(G(\mathbf{z})) - \lambda), \quad (3)$$

$$\max_D \ C_D(D, G; \lambda) = V_{\text{GAN}}(D, G) - L_{\text{regularization}}(D, G; \lambda), \quad (4)$$

$$\min_G \ C_G(D, G; \lambda) = \mathbb{E}_{\mathbf{x} \sim p_{\text{data}}(\mathbf{x})}[\log(D(\mathbf{x}))] + \frac{\lambda - 1}{\lambda} \mathbb{E}_{\mathbf{z} \sim p_{\mathbf{z}}(\mathbf{z})}[\log(1 - D(G(\mathbf{z})))]. \quad (5)$$

The discriminator loss $C_D(D, G; \lambda)$ is the original loss $V_{\text{GAN}}(D, G)$ combined with the regularization term $L_{\text{regularization}}(D, G; \lambda)$, and the generator loss function $C_G(D, G; \lambda)$ is the re-scaled version of $V_{\text{GAN}}(D, G)$. In Equation (4)-(5), the goal of $D$ and $G$ is not completely opposite (unlike Equation (1)), making it a mixed-motive game which contains both the competition and the collaboration mechanism. The collaboration is enforced through the regularization term $L_{\text{regularization}}(D, G; \lambda)$, in which the two networks jointly explore distributions spanning from "authentic" to "synthetic", with the balance controlled by the hyperparameter $\lambda$. We denote the optimization problem in Equation (4)-(5) as **C**ollaboration **C**ompetition-based **G**enerative **A**dversarial **N**etwork (CCGAN).

**Global optimality.** The regularization term in Equation (3) has two advantages. First, it encourages the discriminator to map true data to probability $\lambda$. In contrast to the original discriminator, which always attempts to assign a probability 1 to real data, this additional regularization empowers the model to explore the distributions spanning the spectrum from "true" to "half true". The "half true"

---

[2]Since $L_{\text{regularization}}(D, G; \lambda)$ is a collaborative pursuit involving both the discriminator and generator, its impact is manifested with an opposite sign in Equation (2).

distribution reflects the "true" distribution with specific requisites, and is mathematically mapped to probability $\lambda$. Hence, the hyperparameter $\lambda$ is configured to implement the desired level of specific requisites for the generated data. Second, this regularization term facilitates the transformation of the loss function into a divergence metric. The transformation prevents the introduction of additional complexities when searching for an equilibrium point. This outcome leads to the establishment of global optimality, as demonstrated in Theorem 2.

**Theorem 2** (Global optimality of CCGAN). *The optimization problem in Equation (4)-(5) is equivalent to minimizing the Jensen–Shannon divergence $JSD(p_{data}\|\frac{\lambda-1}{\lambda}\cdot p_{gen})$ at optimality. It leads to $p_{gen} = \frac{\lambda}{\lambda-1}\cdot p_{data}$, where $p_{data}$ is true data distribution, $p_{gen}$ the distribution of generated data.*

*Proof.* The training criterion for the discriminator $D$, given a fixed generator $G$, is to maximize the quantity $V_{GAN}(D, G) - L_{\text{regularization}}(D, G; \lambda)$ as

$$\mathbb{E}_{\mathbf{x}\sim p_{\text{data}}(\mathbf{x})}[\log(D(\mathbf{x})) - \log(D(\mathbf{x}) - \lambda)] + \mathbb{E}_{\mathbf{z}\sim p_{\mathbf{z}}(\mathbf{z})}[\log(1 - D(G(\mathbf{z}))) - \log(\lambda - D(G(\mathbf{z})))]$$

$$= \int_{\mathbf{x}} p_{\text{data}}(\mathbf{x})[\log(D(\mathbf{x})) - \log(D(\mathbf{x}) - \lambda)]d\mathbf{x} + \int_{\mathbf{x}} p_{\text{gen}}(\mathbf{x})[\log(1 - D(\mathbf{x})) - \log(\lambda - D(\mathbf{x}))].$$

Then, the optimal discriminator $D_G^*$ is obtained when

$$(\frac{p_{\text{data}}(\mathbf{x})}{D_G^*(\mathbf{x})} - \frac{p_{\text{data}}(\mathbf{x})}{D_G^*(\mathbf{x}) - \lambda}) + (\frac{p_{\text{gen}}(\mathbf{x})}{D_G^*(\mathbf{x}) - 1} - \frac{p_{\text{gen}}(\mathbf{x})}{D_G^*(\mathbf{x}) - \lambda}) = 0 \implies D_G^*(\mathbf{x}) = \frac{p_{\text{data}}(\mathbf{x})}{p_{\text{data}}(\mathbf{x}) + \frac{\lambda-1}{\lambda}\cdot p_{\text{gen}}(\mathbf{x})}.$$

Given the optimal $D_G^*$, the generator is to minimize the quantity in Equation (5) as

$$\min_G \mathbb{E}_{\mathbf{x}\sim p_{\text{data}}(\mathbf{x})}[\log(D_G^*(\mathbf{x}))] + \frac{\lambda-1}{\lambda}\mathbb{E}_{\mathbf{z}\sim p_{\mathbf{z}}(\mathbf{z})}[\log(1 - D_G^*(G(\mathbf{z})))]$$

$$= \mathbb{E}_{\mathbf{x}\sim p_{\text{data}}(\mathbf{x})}[\log \frac{p_{\text{data}}(\mathbf{x})}{p_{\text{data}}(\mathbf{x}) + \frac{\lambda-1}{\lambda}\cdot p_{\text{gen}}(\mathbf{x})}] + \mathbb{E}_{\mathbf{x}\sim \frac{\lambda-1}{\lambda}\cdot p_{\text{gen}}(\mathbf{x})}[\log \frac{\frac{\lambda-1}{\lambda}\cdot p_{\text{gen}}(\mathbf{x})}{p_{\text{data}}(\mathbf{x}) + \frac{\lambda-1}{\lambda}\cdot p_{\text{gen}}(\mathbf{x})}]$$

$$= -\log 4 + 2\cdot \text{JSD}(p_{\text{data}}\|\frac{\lambda-1}{\lambda}\cdot p_{\text{gen}}). \qquad \square$$

**Avoidance of training collapse.** Besides inheriting the theoretical advantage of original GAN, CC-GAN can also numerically alleviate the training collapse issue, which occurs when the generator consistently produces limited and repetitive samples, resulting in a lack of diversity and low-quality generated data. The root cause of this problem in traditional GANs lies in the pure-competition relationship between the generator and discriminator. Their interdependence can lead to a scenario where stagnation in one network hinders improvements in the other. For addressing this challenge, we argue that the regularization term introduced in Equation (3) breaks the mutual dependency between the generator and discriminator, as demonstrated in Theorem 3. This newfound independence provides the networks with additional incentive to improve even if another network is stagnant. The proof of Theorem 3 is in Appendix B.

**Theorem 3** (Avoid training collapse). *Given the generator, the optimal discriminator is obtained as*

$$D^*(\mathbf{x}) = \frac{[\lambda\gamma(p_{data}(\mathbf{x}) + p_{gen}(\mathbf{x})) + \lambda(p_{data}(\mathbf{x}) - p_{gen}(\mathbf{x}))]^2 + \sqrt{4\lambda\gamma p_{data}(\mathbf{x})((1-\gamma)p_{gen}(\mathbf{x}) - \lambda p_{data}(\mathbf{x}))}}{2(1-\gamma)p_{gen}(\mathbf{x}) - 2\lambda p_{data}(\mathbf{x})},$$

*if the regularization adopts $\mathbb{E}_{\mathbf{x}\sim p_{data}(\mathbf{x})}\log(D(\mathbf{x}) - \lambda) + \mathbb{E}_{\mathbf{z}\sim p_{\mathbf{z}}(\mathbf{z})}\log(D(G(\mathbf{z})) - \gamma)$.*

$$\text{(6)}$$

Overall, CCGAN offers three advantages as follows, and is summarized into Algorithm 1.

- It allows the generator and discriminator to work together to fulfill specific requirements of generated samples that lies beyond the authenticity of the "true" data distribution.
- It preserves the equilibrium point as in original GAN. The presence of an equilibrium point can guide the selection of hyperparameter $\lambda$ and bolster convergence, facilitating the generation of consistently high-quality samples.
- It empowers the generator and discriminator to explore a more extensive parameter space, effectively mitigating the issue of training collapse.

## 4 EXPERIMENTS

In this section we evaluate the advancements of CCGAN, exploring in particular its ability of generating consistently satisfactory samples and avoiding training collapse issue. The experiments were conducted on a single NVIDIA GPU.

---

**Algorithm 1** CCGAN: Collaboration Competition-based Generative Adversarial Network

---

**Input:** Dataset $\mathcal{D} = \{\mathbf{x}^n\}_{n=1}^N$.
**Initialize:** Hyperparameter $\lambda$, generator training gap $k$, number of iterations $T$, batch size $n_0$.
**for** $T$ training iterations **do**
    **for** $k$ steps **do**
        Sample minibatch of $n_0$ images from $\mathcal{D} : \{\mathbf{x}^{(1)}, \cdots, \mathbf{x}^{(n_0)}\}$.
        Sample minibatch of $n_0$ noises from Gaussian distribution : $\{\mathbf{z}^{(1)}, \cdots, \mathbf{z}^{(n_0)}\}$.
        Update the discriminator $D(\cdot)$ by ascending its stochastic gradient:

$$\nabla_D \frac{1}{n_0} \sum_{i=1}^{n_0} \Big[ \log D(\mathbf{x}^{(i)}) - \log[D(\mathbf{x}^{(i)}) - \lambda] + \log\Big(1 - D\big(G(\mathbf{z}^{(i)})\big)\Big) - \log\Big(\lambda - D\big(G(\mathbf{z}^{(i)})\big)\Big) \Big].$$

    Update the generator $G(\cdot)$ by descending its stochastic gradient:

$$\nabla_G \frac{1}{n_0} \sum_{i=1}^{n_0} \Big[ \log\Big(1 - D\big(G(\mathbf{z}^{(i)})\big)\Big) \Big]$$

**Output:** CCGAN with the ability to generate customized images.

---

## 4.1 SETTINGS

**Datasets and Applications.** To assess CCGAN's ability in handling diverse data types, we use four image datasets and one power dataset, each with specific collaboration requirements. The image datasets cover various subjects, including humans, animals, buildings, and natural landscapes. (1) **Coloring grayscale shapes**. We start from simple objects like rectangles and circles. For this, we synthesize a dataset **Syn** containing grayscale images of standardized geometric shapes, where the task is applying suitable colors for different shapes. (2) **Coloring grayscale images**. Then, we proceed to colorize grayscale images of common objects. For this, we choose the Common Objects in Context (**COCO**) dataset Lin et al. (2014). (3) **Image super resolution**. Besides coloring, we consider the task of producing a high-resolution image from its low-resolution counterpart, using the dataset **SuperRes** Bevilacqua et al. (2012); Zeyde et al. (2012) containing life-scenes and natural landscapes. (4) **Multi-modal generation**. We explore multi-modal generation, aiming to create coherent and realistic outputs based on the input information. We use the **Facade** dataset Tyleček & Šára (2013); Isola et al. (2017) and **deepFashion** dataset Liu et al. (2016); Zhu et al. (2020b), which provides paired inputs/outputs for building facades and human clothes, respectively. (5) **False data injection attack**. Besides image data, we study power data from power systems, specifically for an adversarial attack task. It conducts imperceptible (for power utility) modifications to power measurements using the dataset **FDIA** Costilla-Enriquez & Weng (2021).

**Benchmark methods**. We consider baselines in GAN that implement a regularization to promote collaboration between the generator and discriminator. The following methods are utilized: (1) the original GAN (**GAN**) Goodfellow et al. (2014; 2020), (2) conditional coordinating GAN (**COCO-GAN**) Lin et al. (2019), (3) collaborative GAN (**CollaGAN**) Lee et al. (2019), and (4) cooperation GAN (**Co-GAN**) Zhang et al. (2020).

**Implementing details of CCGAN.** For the generator and discriminator architecture, we employ five convolutional layers and two fully-connected layers that incorporates a skip connection He et al. (2016). Each layer comprises approximately ten neurons, and the neurons are activated through pooling layers and Rectified Linear Units (ReLU) Agarap (2018). We set the number of maximal iterations to $T = 300$ for sufficient training. Additionally, for every $k = 5$ iterations will we train the generator so that we prioritize training the discriminator to allow for better convergence. For each iteration, we sample $n_0 = 50$ mini-batches to compute gradients for advanced searching for parameters. We update these parameters using the Adam optimizer with a learning rate of $2 \times 10^{-4}$.

## 4.2 CHALLENGES: NO PERFORMANCE GUARANTEE AND TRAINING COLLAPSE ISSUE

Prior to evaluating CCGAN's improvements, we show two challenges that may arise in existing GAN models. First, the lack of performance guarantee may hinder a successful joint optimization of competition and collaboration in GAN. Specifically, an inconsistent regularization term $L_{\text{regularization}}(D, G; \lambda)$ can result in poor collaboration, leading to the generation of subpar outcomes.

For instance, in image super-resolution (see Figure 2 (a)) and image colorization (see Figure 2 (b)), an improper hyperparameter ($\lambda$) selection to balance competition and collaboration is shown in the third row and indicated by blue arrows. It yields unexpected and unsatisfactory results in both tasks.

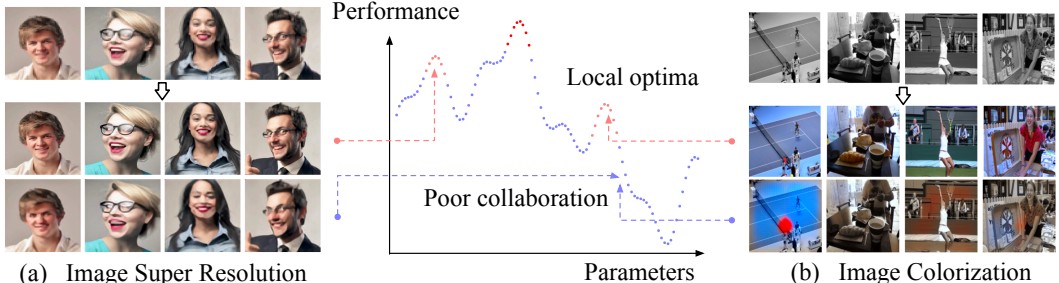

(a) Image Super Resolution          (b) Image Colorization

Figure 2: Two challenges in existing GANs: poor collaboration (blue arrows) and local optima (pink arrows), are simulated in the task of (a) image super resolution and (b) image colorization.

Second, in pure-competition GANs, the mutual dependence between the generator and discriminator often leads to a susceptibility to getting trapped in local optima, a phenomenon known as training collapse. This occurs when the networks reach a state of local optima and lose the incentive to further improve due to their competitive mechanism. For instance, in Figure 2, the seemingly good results in the second row might represent local optima resulting from insufficient training of the two networks, as indicated by the pink arrows.

### 4.3 EVALUATION OF CCGAN

This section evaluates the ability of **CCGAN** to resolve the above two challenges. For this purpose, we provide qualitative and quantitative results as follows.

**The benefits of performance guarantee.** In the colorization task using the **COCO** dataset Lin et al. (2014), we verify the benefit of CCGAN's equilibrium point result as outlined in Theorem 2. To do this, we colorize images to a predefined color spectrum, specifically the yellow and blue channels. This prior knowledge allows us to optimize the hyperparameter $\lambda$ in the regularization term $L_{\text{regularization}}(D, G; \lambda)$, avoiding the need for random selection and post-training observation as in other baselines. As depicted in Figure 3, the baseline model **CollaGAN** may achieve good results after exploring the hyperparameter space, but it remains susceptible to suboptimal hyperparameter choices that lead to unexpected outcomes. In contrast, CCGAN is equipped with optimized hyperparameter due to the existence of equilibrium point. Thus, it can accurately infer and apply pre-defined colors to grayscale inputs. This eliminates the troublesome process of hyperparameter tuning and the associated risk of selecting suboptimal values.

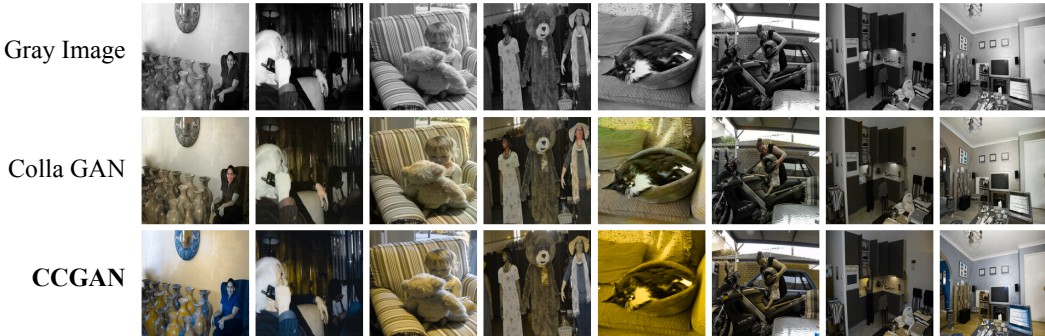

Figure 3: Image colorization: CCGAN leads to optimal with equilibrium point in Theorem 2.

**The avoidance of training collapse.** To evaluate CCGAN's efficacy in mitigating training collapse, we consider the multi-modal generation task. This task involves generating a new image that is realistic and coherent with an input image, necessitating the model to capture multiple modes within the underlying distribution. To achieve this, GAN models often incorporate regularization terms, striking a balance between fidelity and diversity. However, regularization can also lead to training collapse when applied inconsistently, resulting in unrealistic or incoherent generated images. The

results of multi-modal generation on both the **Facade** dataset Tyleček & Šára (2013); Isola et al. (2017) and **deepFashion** dataset Liu et al. (2016); Zhu et al. (2020b) are shown in Figure 4.

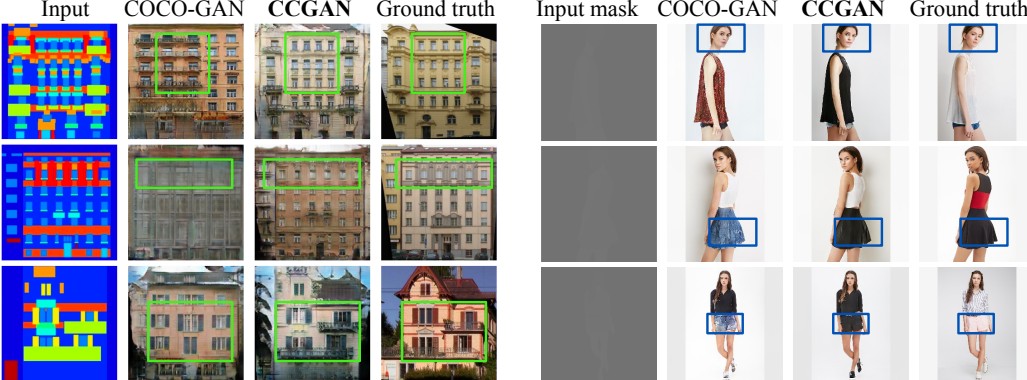

(a) : Image-to-image generation: buildings    (b) : Image-to-image generation: human clothes

Figure 4: The multi-modal reconstruction results of CCGAN and baseline method.

As depicted in Figure 4, the images produced by CCGAN exhibit superior clarity and coherence when contrasted with those generated by COCO-GAN. This distinction may stem from the inherent nature of the regularization loss within our design, which allows the model to explore more modes in the data distribution, as discussed in Theorem 3. Furthermore, our model benefits from the performance guarantee established in Theorem 2, allowing it to achieve more precise reconstructions with smoothly delineate image edges.

**Enhanced performance of CCGAN with regular-shaped objects.** After evaluating CCGAN's capabilities, we want to identify the scenarios where CCGAN exhibits enhanced efficacy. We hypothesize that CCGAN, benefiting from its equilibrium point, thrives when processing images containing "convex" objects. These objects can be represented by convex functions within neural networks, allowing models with global optimality guarantee to excel. To test our hypothesis, we created a synthetic dataset **Syn** comprising various geometric shapes, from regular rectangles to irregular circles. We argue that rectangles, describable by multiple line segments, exhibit "convexity" more than irregular circles. In Figure 5, we compare **CCGAN** and **Co-GAN** in applying colors to these synthetic geometric shapes. The results reveal that Co-GAN struggles more with rectangles than circles. This may be attributed to inconsistent regularization, leading to unclear and jagged edges in generated images, thereby diminishing their quality and realism. In contrast, our model, benefiting from the equilibrium point, excels at discerning objects with well-defined shapes and edges.

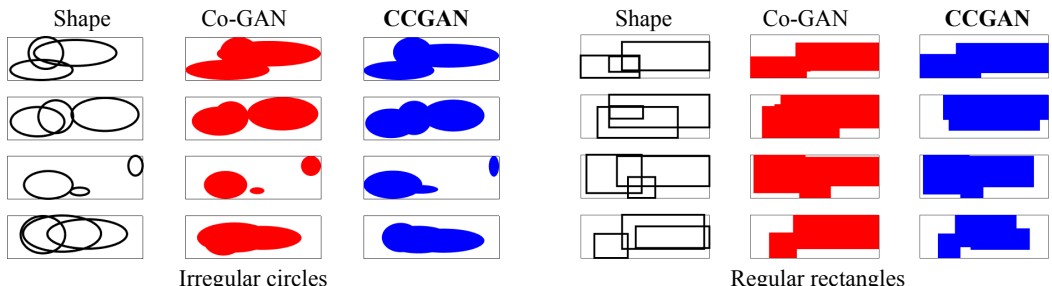

Irregular circles    Regular rectangles

Figure 5: Image colorization on synthetic shapes.

In addition to testing this phenomenon with synthetic data, we extend our validation to real-world datasets, specifically in the context of image super-resolution task. This task often incorporate regularization into GAN models to faithfully reproduce the low-resolution input. Figure 6 presents visual results comparing **CCGAN** and baselines. The comparison highlights CCGAN's ability to produce clearer, higher-resolution reconstructions. Notably, CCGAN exhibits superior performance when processing objects with regular shapes and well-defined edges. This heightened performance can be attributed to the inherent performance guarantee of the CCGAN design, which promotes greater accuracy when the depicted objects in the image exhibit convex characteristics.

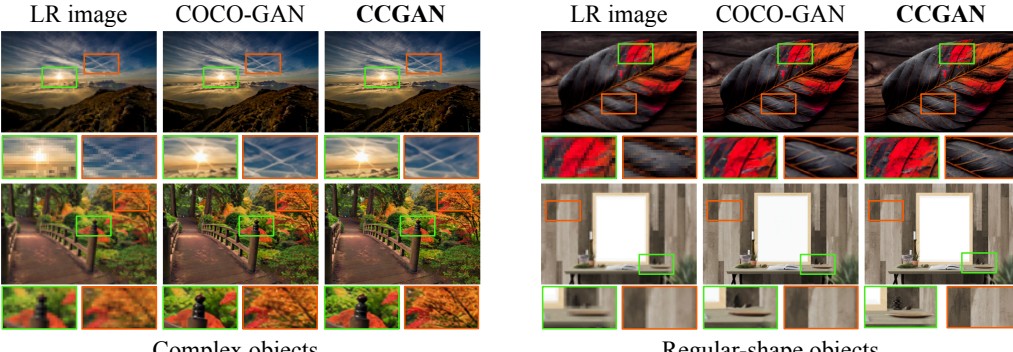

Figure 6: CCGAN and COCO-GAN reconstruction results and reference low-resolution (LR) image.

For image super-resolution task, we also evaluate the result using commonly used metrics, including peak signal-to-noise ratio (PSNR) Yang et al. (2007), structural similarity (SSIM) Wang et al. (2004) and mean opinion score (MOS) . The results in Table 1 validate our performance is better than the baselines, verifying the capability of CCGAN to generate satisfactory data catering to specific need.

Table 1: Comparison of CCGAN and baselines and the original High Resolution (SR) image.

| SuperRes | COCO-GAN | CollaGAN | Co-GAN | CCGAN | HR |
|----------|----------|----------|--------|-------|-----|
| PSNR ↑ | 29.53 | 28.91 | 29.40 | **30.43** | $\infty$ |
| SSIM ↑ | 0.8621 | 0.8835 | 0.8472 | **0.9011** | 1 |
| MOS ↑ | 3.19 | 2.85 | 3.64 | **3.69** | 4.25 |

**Performance of CCGAN on generating power data.** Beyond image datasets, we explore power data from power systems. We consider a scenario known as the false data injection attack. This task assesses CCGAN's ability to create deceptive power data based on real power data, capable of passing the chi-square test for power utility evaluation. Notably, power data differs from image data, representing continuous time-series data, which enriches our validation on CCGAN. Figure 7 displays CCGAN's superior performance in generating deceptive data based on the **FDIA** dataset, as it yields a higher probability of passing the chi-square test compared to baseline models.

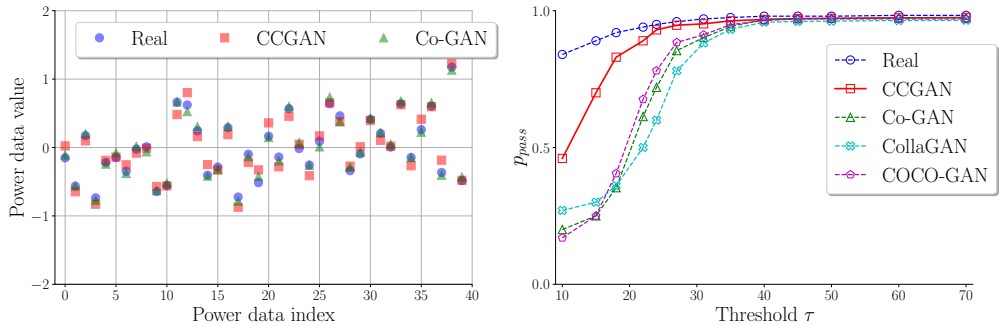

Figure 7: False data injection results from CCGAN and baselines

## 5 CONCLUSION

We presente a technique that enables collaboration in Generative Adversarial Networks. Unlike prior work, our proposed framework preserves the performance guarantee of original GAN, and mitigates the training collapse issue. In specific, we introduced an elegant transformation from the collaboration regularization term to a distribution divergence metric, avoiding addition complexity when seeking an equilibrium point. Our work not only enhances collaboration, interpretability, and the overall optimality of GAN models, but also paves the way for exploring generative models capable of meeting additional requirements with a performance guarantee.

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

## A    PROOF OF THEOREM 1

*Proof.* The formulation in Equation (2) can be seen as the following optimization over the generator and discriminator:

$$\min_{p_{\text{gen}}} \underbrace{\text{JS}(p_{\text{data}}, p_{\text{gen}})}_{\text{learn "true" distribution}} + \lambda \cdot \underbrace{L_{\text{regularization}}(p_{\text{need}}, p_{\text{gen}})}_{\text{fulfill "other" requirements}}, \tag{7}$$

where the original loss $V_{\text{GAN}}(D, G)$ is equivalent Goodfellow et al. (2014) to the Jensen–Shannon (JS) divergence $\text{JS}(p_{\text{data}}\|p_{\text{gen}})$ between true data distribution $p_{\text{data}}$ and generated data distribution $p_{\text{gen}}$. Minimizing the JS divergence $\text{JS}(p_{\text{data}}\|p_{\text{gen}})$ can make learned distribution $p_{\text{gen}}$ match the true distribution $p_{\text{gen}}$. Minimizing the $L_{\text{regularization}}$ fulfills specific need $p_{\text{need}}$ beyond the authenticity of the data distribution. Under the assumption of Gaussianality, we denote $p_{\text{data}} = \mathcal{N}(\mu_1, \sigma_1^2)$, $p_{\text{need}} = \mathcal{N}(\mu_2, \sigma_2^2)$ and $p_{\text{gen}} = \mathcal{N}(\mu, \sigma^2)$. Then, Equation (7) reads as

$$\min_{\mu,\sigma} \text{JS}(\mathcal{N}(\mu_1, \sigma_1^2), \mathcal{N}(\mu, \sigma^2)) + \lambda \cdot L_{\text{regularization}}(\mathcal{N}(\mu_2, \sigma_2^2), \mathcal{N}(\mu, \sigma^2))$$

$$= \min_{\mu,\sigma} \log \frac{\sigma_1^2 + \sigma^2}{2\sigma_1\sigma} + \frac{(\mu_1 - \mu)^2}{\sigma_1^2 + \sigma^2} + \lambda \cdot (\mu_2 - \mu)^2 + \lambda \cdot (\sigma_2^2 - \sigma^2)^2. \tag{8}$$

For deriving the equilibrium point of $(\mu, \sigma)$, we assign corresponding derivative for $\mu$ to zero as

$$\frac{2(\mu_1 - \mu)}{\sigma_1^2 + \sigma^2} + 2\lambda \cdot (\mu_2 - \mu) = 0 \Rightarrow \mu = \frac{\mu_1 + \lambda(\sigma_1^2 + \sigma^2)\mu_2}{1 + \lambda(\sigma_1^2 + \sigma^2)}. \tag{9}$$

Taking the above result into Equation (8) and assign derivative for $\sigma$ to zero, we have

$$\frac{\sigma^2 - \sigma_1^2}{(\sigma^2 + \sigma_1^2)\sigma} - 4\lambda\sigma(\sigma_2^2 - \sigma^2) + \lambda(\mu_2 - \mu_1)^2 \frac{-2\lambda\sigma}{[1 + (\sigma^2 + \sigma_1^2)]^2} = 0, \tag{10}$$

which is a polynomial equation of degree higher than five, where an analytic (closed-form) solution is absent Ramond (2022). □

## B    PROOF OF THEOREM 3

*Proof.* The training criterion for the discriminator $D$, given a fixed generator $G$, is to maximize the quantity $V_{\text{GAN}}(D, G) - L_{\text{regularization}}(D, G; \lambda)$ as

$$\mathbb{E}_{\mathbf{x} \sim p_{\text{data}}(\mathbf{x})}[\log(D(\mathbf{x})) - \log(D(\mathbf{x}) - \lambda)] + \mathbb{E}_{\mathbf{z} \sim p_{\mathbf{z}}(\mathbf{z})}[\log(1 - D(G(\mathbf{z}))) - \log(\gamma - D(G(\mathbf{z})))]$$

$$= \int_{\mathbf{x}} p_{\text{data}}(\mathbf{x})[\log(D(\mathbf{x})) - \log(D(\mathbf{x}) - \lambda)]d\mathbf{x} + \int_{\mathbf{z}} p_{\mathbf{z}}(\mathbf{z})[\log(1 - D(G(\mathbf{z}))) - \log(\gamma - D(G(\mathbf{z})))]$$

$$= \int_{\mathbf{x}} p_{\text{data}}(\mathbf{x})[\log(D(\mathbf{x})) - \log(D(\mathbf{x}) - \lambda)]d\mathbf{x} + \int_{\mathbf{x}} p_{\text{gen}}(\mathbf{x})[\log(1 - D(\mathbf{x})) - \log(\gamma - D(\mathbf{x}))].$$

Then, the optimal discriminator $D_G^*$ is obtained when

$$(\frac{p_{\text{data}}(\mathbf{x})}{D_G^*(\mathbf{x})} - \frac{p_{\text{data}}(\mathbf{x})}{D_G^*(\mathbf{x}) - \lambda}) + (\frac{p_{\text{gen}}(\mathbf{x})}{D_G^*(\mathbf{x}) - 1} - \frac{p_{\text{gen}}(\mathbf{x})}{D_G^*(\mathbf{x}) - \gamma}) = 0$$

$$\implies -\lambda p_{\text{data}}(\mathbf{x})(D_G^*(\mathbf{x}) - 1)(D_G^*(\mathbf{x}) - \gamma) + (1 - \gamma)p_{\text{gen}}(\mathbf{x})D_G^*(\mathbf{x})(D_G^*(\mathbf{x}) - \lambda)$$

$$\implies D_G^*(\mathbf{x}) = \frac{1}{2}\frac{[\lambda\gamma(p_{\text{data}}(\mathbf{x}) + p_{\text{gen}}(\mathbf{x})) + \lambda(p_{\text{data}}(\mathbf{x}) - p_{\text{gen}}(\mathbf{x}))]^2 + \sqrt{4\lambda\gamma p_{\text{data}}(\mathbf{x})((1 - \gamma)p_{\text{gen}}(\mathbf{x}) - \lambda p_{\text{data}}(\mathbf{x}))}}{(1 - \gamma)p_{\text{gen}}(\mathbf{x}) - \lambda p_{\text{data}}(\mathbf{x})}.$$

□

