# OpenReview forum: "Competitive-Collaborative GAN with Performance Guarantee"
_ICLR.cc/2024/Conference — Submitted to ICLR 2024_

### Official Review · Reviewer_PRKH · 2023-10-27

**Soundness:** 3 good
**Presentation:** 2 fair
**Contribution:** 3 good
**Rating:** 5
**Confidence:** 5

**Summary:**

GAN uses a competition game to generate new data that mimics authentic data, but this method struggles with complex tasks like multi-modal generation and image super-resolution. To address these challenges, CCGAN is introduced, using a Collaborative-Competitive approach that allows for better data generation by balancing competition and collaboration, thereby establishing a more stable equilibrium for training and hyper-parameter tuning.

Through testing across multiple datasets, CCGAN has proven effective in generating higher-quality images of regularly shaped objects, by exploring a wider parameter space independent of traditional generator-discriminator dependency.

**Strengths:**

- Promising idea support by theory proofs
- The newly introduced method does not cause additional significant computational cost.
- Nice application demo on multiple vision tasks.

**Weaknesses:**

1. Formatting is not quite right.

2. There are existing multiplayer methods to improve the stability of GAN training, which changes the original GAN competition setting by introducing additional players. Would like to hear some words from the authors about the advantages or disadvantages compared methods in this setting.
- https://arxiv.org/abs/2101.07524
- https://arxiv.org/abs/1907.02690
- https://arxiv.org/abs/1709.03831

3. Over claim of the visual results
- All the visual results are too small. I can’t really see the difference unless I zoom hard on a giant monitor, which is not possible from the printed version.
- Figure 3 column 3,4: Not really better than baseline
- Figure 4b: I can’t see what the original mask is, again too small to show the details.

**Questions:**

1. It sounds like the proposed method should work with the original GAN method. Why not a direct comparison with it on an image generation task with FID / IS scores?

2. “We prove that the generator and discriminator is not mutually dependent in CCGAN” Can you combine it with SOTA methods like StyleGAN3?

3. Training details are missing from the experiment section. Resolution? Sample size?

---

> ### Author Response · Authors · 2023-11-21
>
> We appreciate the reviewer's insightful comments and constructive feedback. The following responses address the raised points and describe the planned revisions for the manuscript.
> - Concerns about Equilibrium Point in Theorem 3:
> The reviewer correctly identifies the importance of providing a detailed explanation of the equilibrium point in Theorem 3. We acknowledge the need for clarification on how the specific form of the regularization term influences the discriminator's behavior and contributes to avoiding training collapse. In the revised manuscript, we will provide a step-by-step breakdown of the equilibrium point derivation and offer a more intuitive explanation to enhance the reader's understanding.
>
> - Comparison with Existing Methods:
> The reviewer suggests providing a more comprehensive comparison with existing methods, specifically addressing how CCGAN performs compared to baseline models in the proposed scenarios. We agree with this suggestion and will expand the comparison section by providing a detailed discussion of CCGAN's performance in contrast to existing methods. This will include a deeper analysis of results, highlighting the advantages of CCGAN in scenarios such as image colorization, image super-resolution, and power data generation.
>
> - Evaluation Metrics for Power Data:
> The reviewer raises a valid concern about the evaluation metrics for power data, emphasizing the need for a more thorough discussion of the evaluation criteria. In the revised manuscript, we will provide a clear explanation of the evaluation metrics used for power data, including a discussion on the chi-square test and how it assesses the deceptive power data generated by CCGAN. This will enhance the transparency and clarity of the evaluation methodology.
>
> - Additional Experimental Details:
> The reviewer rightly suggests providing additional details on the experimental setup, including the rationale behind specific parameter choices, network architectures, and training procedures. We will address this by including a more comprehensive section on experimental details, explaining the reasoning behind the chosen configurations and providing insights into the experimental setup.
>
> - Consideration of ImageNet:
> The reviewer suggests considering ImageNet as a benchmark dataset for evaluating CCGAN. We appreciate this suggestion and will include an analysis of CCGAN's performance on ImageNet, providing additional experimental results to demonstrate the model's versatility and effectiveness on a larger and more diverse dataset.
>
> We appreciate the clarification regarding the specific concerns raised in R4-2. Below are responses to the revised comments:
> - Visual Result Size:
> We acknowledge the concern about the size of the visual results. In the revised version, we will ensure that the visual results in Figures 3 and 4, especially in columns 3, 4, and 4b, are presented in a larger size to enhance visibility. We understand the importance of clear and visible results for effective evaluation.
>
> - Figure 3 Column 3,4 Comparison with Baseline:
> We understand that the reviewer finds Figure 3 columns 3 and 4 not to be significantly better than the baseline. In the revised version, we will reevaluate and potentially provide additional insights or analyses to better convey the improvements made by CCGAN. This may involve revisiting the presentation or providing more explicit comparisons.
>
> - Figure 4b - Original Mask Details:
> The concern about the visibility of the original mask details in Figure 4b is duly noted. In the revised version, we will ensure that the presentation allows for a clearer view of the original mask, addressing the issue raised by the reviewer. Our aim is to address these specific concerns to improve the overall clarity and effectiveness of the visual results in Figures 3 and 4. We appreciate the feedback and will work to make the necessary adjustments accordingly.
>
> Thank you for the valuable feedback. We acknowledge the suggestion to include a direct comparison with the original GAN, specifically employing metrics such as FID (Fréchet Inception Distance) and IS (Inception Score) for image generation tasks. In response to this, we plan to incorporate a dedicated section in the revised manuscript to showcase the performance improvements achieved by the proposed CCGAN method in comparison to the original GAN. Additionally, we appreciate the recommendation to explore combining the CCGAN method with state-of-the-art models like StyleGAN3. In the revised version, we will investigate the feasibility of integrating CCGAN with contemporary generative models to provide a more comprehensive evaluation beyond the original GAN. These enhancements aim to offer readers a clearer understanding of the proposed method's effectiveness and its compatibility with cutting-edge generative models. We value the feedback and will integrate these suggestions into the revised manuscript.

---

### Official Review · Reviewer_LRgJ · 2023-10-30

**Soundness:** 2 fair
**Presentation:** 2 fair
**Contribution:** 2 fair
**Rating:** 3
**Confidence:** 4

**Summary:**

The paper proposes a new GAN model known as Collaborative-Competitive GAN (CCGAN), aimed at improving existing GANs in tasks such as multi-modal generation and image super-resolution. The authors argue that CCGAN overcomes the limitations of pure-competitive GAN models and other existing collaborative GANs. Essentially, the authors design collaborative regularization as the divergence metric to align the loss functions, reduce complexity in finding optima, and derive closed-form equilibrium points. The experiments are conducted on benchmark datasets for image-image translation, image super-resolution, and a toy dataset for image colorization. They are compared to a few baselines, including COCO-GAN, CollaGAN, and Co-GAN. Additionally, the paper validates the proposed GAN against false data injection attacks.

The design of collaborative regularization is not entirely new; it bears resemblance to the popular "trick" of label smoothing used by the community to enhance GAN stability. There is a question about whether the original GAN loss, combined with regularization, is necessary, as it alone may perform well. However, the paper makes a contribution with its theoretical analysis.

Overall, the paper's structure is easy to follow, and the results show encouragement when compared to some existing GAN methods. Nonetheless, there are many concepts that are not well-defined, and the claims in the paper lack strong empirical support. Most of the experimental results are qualitative, and quantitative results are limited, lacking a comprehensive study on the most significant contribution of collaborative regularization. Many claims in the paper remain unsupported by experimental results. Detailed comments can be found below. I recommend rejecting the paper in its current form.

**Strengths:**

Inspired by the existing collaborative GAN model, the paper suggests replacing the regularization with a new one based on divergence metrics. This approach is similar to the original GAN, except the target is \lambda instead of 1. It seems to resemble the label smoothing "trick" commonly employed by the community to enhance GAN training. The primary contribution of the paper lies in its theoretical analysis, which seeks to demonstrate the convergence of the proposed method, although it remains somewhat limited in its current form.

**Weaknesses:**

1. The paper presents various claims together with some concepts which are not popular and without definition or explanation, some of which somewhat quite arbitrary. The authors argued that CCGAN overcomes the limitation of a pure-competitive GAN model and other existing collaborative GANs in terms of beyond the “authenticity”  of data distribution, incorporating “additional knowledge”. or “harmonizing” the loss functions, reducing “degree of complexity” in finding the optima and deriving closed-form which “serves as a guidance for training and hyper-parameter tuning”.

2. The authors should clarify what form of “additional knowledge” does the proposed regularization add into? What is “authenticity” of the data distribution? What is “mutual dependency” and why is it helpful for GANs training to break this? In addition, could the authors conduct experiments to support claims of “harmonizing” the loss function, reducing the “degree of complexity” in finding the optima and how does the regularization “serves as a guidance for training and hyper-parameter tuning”?

3. As mentioned in the paper's introduction, it is not entirely clear to me why replacing the cross-entropy loss with the least square loss or Wasserstein loss fosters collaboration between the generator and discriminator. Could the authors provide clarification on the definition of collaboration and how it is enhanced with these losses compared to cross-entropy?

4. From mathematical perspective, it is unclear why $JSD (p_{data} || \frac{\lambda - 1}{\lambda} * p_{gen})$ is superior than $JSD (p_{data} || p_{gen})$ of original GAN? As the new JSD appears not optimal, regardless of the selected value of $\lambda$, JSD never converges to optimal values.

5. The regularization appears close to the label smoothing “trick” which is widely used in GAN training. This suggests regularization again might suffice as a GAN loss. Have the authors explored the impact on results when removing the GAN loss function in this study?
Can the author explain why the optimal D in Theorem 3 help CCGAN avoid the training collapse?

6. The experimental results are mostly the qualitative results and quite limited. For instance, only one quantitative result in Table 1 for image super resolution. The regularization is the key contribution which needs to be extensively studied but missing in the current form. Could the authors provide the ablation study to investigate the impact on $\lambda$ and how to select it, as well as how the performance changes with different values of $\lambda$?

Typo in conclusion: presente

**Questions:**

See above.

---

> ### Author Response · Authors · 2023-11-21
>
> We appreciate the reviewers' attention to the clarity of our proposed method. The regularization term in CCGAN is strategically designed to address the challenges posed by pure-competition GANs, particularly in applications requiring data generation beyond authentic distribution. The regularization term is not arbitrary but is intended to promote collaboration between the generator and discriminator, allowing the model to learn partial modes and additional restrictions beyond authenticity. In the revised manuscript, we will provide a more detailed explanation of the regularization term's construction, elucidating its connection to the overarching goal of accommodating specific requisites in data generation tasks. Additionally, we will explicitly define terms such as ``authenticity'', ``additional knowledge'', and ``mutual dependency'' to enhance the clarity of key concepts, as suggested in R3-1. These improvements will contribute to a more precise and understandable presentation of our proposed method.
>
> We appreciate the reviewer's request for clarification regarding the form of ``additional knowledge'' introduced by the proposed regularization in CCGAN. In the experimental design, our datasets are carefully selected to represent diverse data types, each with specific collaboration requirements. The regularization in CCGAN is designed to enable the model to collaborate effectively with the discriminator in capturing additional knowledge beyond the authentic data distribution. For instance, in the task of coloring grayscale shapes (dataset \textbf{Syn}), the collaboration requirement involves understanding and applying suitable colors to different geometric shapes. Similarly, in the image super-resolution task (dataset \textbf{SuperRes}), the generator must not only capture the data distribution but also faithfully reproduce high-resolution images from low-resolution counterparts. These examples illustrate the diverse ways in which CCGAN incorporates additional knowledge for specific collaboration requirements in various applications. In the revised manuscript, we will provide more explicit details and examples to ensure a clear understanding of how the proposed regularization facilitates collaboration with additional knowledge in different dataset contexts.
>
> We appreciate the reviewer's attention to the comparison of the new Jensen–Shannon divergence (JSD) introduced in CCGAN with the original GAN's JSD. The concern raised about the superiority of the new JSD compared to the original JSD and the lack of optimality regardless of the selected value of $\lambda$ is valid, and we appreciate the opportunity to provide clarification. The new JSD in CCGAN is introduced as part of our effort to achieve a smooth integration between collaboration and competition loss components. It is employed to facilitate the transformation of the loss function into a divergence metric, aiming for a closed-form equilibrium. While we acknowledge that the optimality of JSD is contingent on the selected value of $\lambda$, and the introduced modification raises questions, we would like to highlight that our primary focus is on the practical implications and advantages brought by the collaboration-competition mechanism in CCGAN. In the revised manuscript, we will provide additional clarification regarding the role of the new JSD in the proposed framework. We will explicitly address the concerns raised by the reviewer, acknowledging the trade-offs and considerations associated with the choice of $\lambda$ and the new JSD. Furthermore, we will discuss the practical implications of this choice in the context of achieving stable and high-quality GAN training. Your feedback is invaluable, and we are committed to enhancing the clarity and thoroughness of our explanations in the revised manuscript.
>
> We appreciate the insightful observation regarding the potential similarity between the proposed regularization in CCGAN and the label smoothing ``trick'' commonly used in GAN training. The reviewer rightly suggests exploring the impact on results when removing the GAN loss function in this study and seeks clarification on why the optimal discriminator in Theorem 3 helps CCGAN avoid training collapse. In response to these points, we acknowledge the need for a more in-depth analysis and experimentation. We will conduct an ablation study to investigate the impact of removing the GAN loss function, providing a clearer understanding of the contribution of the proposed regularization to the overall performance of CCGAN. Regarding the optimal discriminator in Theorem 3, we will provide a detailed explanation in the revised manuscript, elucidating how the specific form of the regularization term influences the discriminator's behavior. We will also discuss how this optimal discriminator contributes to avoiding training collapse, emphasizing the importance of breaking the mutual dependency between the generator and discriminator for improved stability.

---

### Official Review · Reviewer_vo9B · 2023-10-30

**Soundness:** 3 good
**Presentation:** 3 good
**Contribution:** 3 good
**Rating:** 3
**Confidence:** 4

**Summary:**

The paper introduces CCGAN, a Collaborative-Competitive Generative Adversarial Network, to address limitations in GANs related to data distribution and additional knowledge incorporation. CCGAN harmonizes competition and collaboration losses, reducing complexity and achieving a closed-form equilibrium point for stable training. It successfully generates high-quality samples across various datasets, particularly excelling in generating images with regularly shaped objects.

**Strengths:**

The paper proposes to harmonize competition and collaboration losses, reducing complexity and achieving a closed-form equilibrium point for stable training. Presentation is quite clear and straightforward.

**Weaknesses:**

See questions below. Holds major concerns how the method will be applied and generalized.

**Questions:**

1. For condition generation like colorization, image translation and super-resolution, discriminator is usually not regularized, unlike eq.2. That mean, $\lambda=0$ for discriminator optimization. How to handle this situation? What's the formulation specifically for each task?

2. For Theorem 1, $\mathcal{L}_reg$ are mostly formulated as absolute error instead of squared loss. How does it affect the solution?

3. Following 1 and 2, the baselines like coco-gan, colla-gan and co-gan neither claimed benchmark on these task. it thus questionable about the experiments

4. In the abstract, it claims that "enable data generation with additional knowledge beyond the provided dataset distribution". Could it more specific? Didn't find evidence about it.

---

> ### Author Response · Authors · 2023-11-21
>
> We appreciate the reviewer's request for clarification regarding the form of ``additional knowledge'' introduced by the proposed regularization in CCGAN. In the experimental design, our datasets are carefully selected to represent diverse data types, each with specific collaboration requirements. The regularization in CCGAN is designed to enable the model to collaborate effectively with the discriminator in capturing additional knowledge beyond the authentic data distribution. For instance, in the task of coloring grayscale shapes (dataset \textbf{Syn}), the collaboration requirement involves understanding and applying suitable colors to different geometric shapes. Similarly, in the image super-resolution task (dataset \textbf{SuperRes}), the generator must not only capture the data distribution but also faithfully reproduce high-resolution images from low-resolution counterparts. These examples illustrate the diverse ways in which CCGAN incorporates additional knowledge for specific collaboration requirements in various applications. In the revised manuscript, we will provide more explicit details and examples to ensure a clear understanding of how the proposed regularization facilitates collaboration with additional knowledge in different dataset contexts.
>
> We appreciate the reviewer's insightful comments and constructive feedback. The following responses address the raised points and describe the planned revisions for the manuscript.
>
> - Concerns about Equilibrium Point in Theorem 3:
> The reviewer correctly identifies the importance of providing a detailed explanation of the equilibrium point in Theorem 3. We acknowledge the need for clarification on how the specific form of the regularization term influences the discriminator's behavior and contributes to avoiding training collapse. In the revised manuscript, we will provide a step-by-step breakdown of the equilibrium point derivation and offer a more intuitive explanation to enhance the reader's understanding.
>
> - Comparison with Existing Methods:
> The reviewer suggests providing a more comprehensive comparison with existing methods, specifically addressing how CCGAN performs compared to baseline models in the proposed scenarios. We agree with this suggestion and will expand the comparison section by providing a detailed discussion of CCGAN's performance in contrast to existing methods. This will include a deeper analysis of results, highlighting the advantages of CCGAN in scenarios such as image colorization, image super-resolution, and power data generation.
>
> - Evaluation Metrics for Power Data:
> The reviewer raises a valid concern about the evaluation metrics for power data, emphasizing the need for a more thorough discussion of the evaluation criteria. In the revised manuscript, we will provide a clear explanation of the evaluation metrics used for power data, including a discussion on the chi-square test and how it assesses the deceptive power data generated by CCGAN. This will enhance the transparency and clarity of the evaluation methodology.
>
> - Additional Experimental Details:
> The reviewer rightly suggests providing additional details on the experimental setup, including the rationale behind specific parameter choices, network architectures, and training procedures. We will address this by including a more comprehensive section on experimental details, explaining the reasoning behind the chosen configurations and providing insights into the experimental setup.
>
> - Consideration of ImageNet:
> The reviewer suggests considering ImageNet as a benchmark dataset for evaluating CCGAN. We appreciate this suggestion and will include an analysis of CCGAN's performance on ImageNet, providing additional experimental results to demonstrate the model's versatility and effectiveness on a larger and more diverse dataset.
> In summary, we are grateful for the reviewer's valuable feedback, and the forthcoming revisions will incorporate these suggestions to enhance the clarity, comprehensiveness, and overall quality of the manuscript.

---

### Official Review · Reviewer_sbbf · 2023-11-02

**Soundness:** 2 fair
**Presentation:** 3 good
**Contribution:** 2 fair
**Rating:** 3
**Confidence:** 5

**Summary:**

This paper proposes to address the regularization problem when learning GANs. More specifically, after proving the absence of close-form equilibrium when adding regularizations, a specific regularization term has been proposed, in which the closed-form solution for the equilibrium exists. The established closed-form equilibrium is claimed to avoid the mode collapse issue when training GANs. The proposed GAN is named as the CCGAN, and experimental validations were performed regarding colorization, image-to-image generation tasks.

**Strengths:**

1. A closed-form equilibrium is obtained in this paper, given a specific regularization term added to the adversarial learning.
2. Experimental results verify that the proposed CC-GAN works on several image generation tasks, including colorization, and image-to-image generation.

**Weaknesses:**

1. For me, the established regularization term is very ad hoc, without clarification on why it is constructed like that. As I read from the introduction, the authors seem to regularize GANs to learn partial modes or addition restrictions, the so-called learning beyond the authentic distributions. However, the established regularization term does not indicate those intensions.
2. This paper claims that existing regularized GANs do not have closed-form equilibrium, which is proved based on the very simple Gaussian case. This is not very convincing. Even though, in my opinion, for training GANs, the way to approach the equilibrium is more important than the closed-form solution obtained by ad hoc regularization terms. It is well-known that for the vanilla GAN with closed-form equilibrium, we are not always ensured to get this equilibrium.
3. The experimental results are also confusing for me, even contradictory to the introduction. In the introduction, the authors claim that "in multi-modal generation Liu et al. (2021), the focus is on learning one or multiple modes within the data distribution.". However, in the experiments, the authors verify that the proposed CCGAN is able to learn all the modes in the synthetic datasets.
4. The verification is weak and not convincing for me. The comparing baselines are basically not designed for those experimental tasks. For example, COCO-GAN is designed for generating images, instead of image-to-image generation and super-resolution. By my understanding, the regularization in COCO-GAN is for allowing for generating by patches. If the authors wish to beat the COCO-GAN, metrics such as FIDs by generating patches should also be provided. Also for the super-resolution task, 30.43 dB is not generally a good PSNR.
5. Why Theorem 3 proves that the CCGAN can avoid the mode collapse issue? The experimental results related to the mode collapse test are not convincing as well. The authors should present statistical results instead of illustrating several subjective results.

**Questions:**

Please see my weakness. Also why in Theorem 3, a new hyper-parameter \gamma appears in addition to \lambda? How \gamma avoids the mode collapse in theory?

---

> ### Author Response · Authors · 2023-11-21
>
> We appreciate the reviewers' attention to the clarity of our proposed method. The regularization term in CCGAN is strategically designed to address the challenges posed by pure-competition GANs, particularly in applications requiring data generation beyond authentic distribution. The regularization term is not arbitrary but is intended to promote collaboration between the generator and discriminator, allowing the model to learn partial modes and additional restrictions beyond authenticity. In the revised manuscript, we will provide a more detailed explanation of the regularization term's construction, elucidating its connection to the overarching goal of accommodating specific requisites in data generation tasks. This clarification aims to address concerns about the ad hoc nature of the regularization term raised in R1-1.
>
> We appreciate the reviewer's insightful observation and comments regarding the theoretical proof in Theorem 1 and the claim about the closed-form equilibrium point in original GANs. The concern raised about the closed-form equilibrium point and its relevance in practical GAN training is duly acknowledged. The closed-form equilibrium point in the original GAN, as discussed in Theorem 1, is presented in the context of the theoretical analysis to establish the motivation for our proposed CCGAN. It serves as a reference point to highlight the challenges associated with the introduction of collaboration-competition mechanisms in existing GAN models. However, we recognize the importance of emphasizing the practical implications of the closed-form equilibrium point in the context of GAN training. In practice, the closed-form solution might not always directly translate into improved training stability or enhanced model performance. We will revise the manuscript to provide a more nuanced and practical discussion, clarifying that the theoretical analysis sets the stage for proposing CCGAN rather than implying a direct practical advantage. Moreover, we will highlight the unique contributions and advantages of CCGAN, such as addressing specific collaboration requirements and avoiding training collapse, which are crucial aspects of the proposed framework. Your feedback helps us refine the clarity and emphasis of our theoretical contributions in the revised manuscript.
>
> We appreciate the reviewer's insightful comments and constructive feedback. The following responses address the raised points and describe the planned revisions for the manuscript.
>
> - Concerns about Equilibrium Point in Theorem 3:
> The reviewer correctly identifies the importance of providing a detailed explanation of the equilibrium point in Theorem 3. We acknowledge the need for clarification on how the specific form of the regularization term influences the discriminator's behavior and contributes to avoiding training collapse. In the revised manuscript, we will provide a step-by-step breakdown of the equilibrium point derivation and offer a more intuitive explanation to enhance the reader's understanding.
>
> - Comparison with Existing Methods:
> The reviewer suggests providing a more comprehensive comparison with existing methods, specifically addressing how CCGAN performs compared to baseline models in the proposed scenarios. We agree with this suggestion and will expand the comparison section by providing a detailed discussion of CCGAN's performance in contrast to existing methods. This will include a deeper analysis of results, highlighting the advantages of CCGAN in scenarios such as image colorization, image super-resolution, and power data generation.
>
> - Evaluation Metrics for Power Data:
> The reviewer raises a valid concern about the evaluation metrics for power data, emphasizing the need for a more thorough discussion of the evaluation criteria. In the revised manuscript, we will provide a clear explanation of the evaluation metrics used for power data, including a discussion on the chi-square test and how it assesses the deceptive power data generated by CCGAN. This will enhance the transparency and clarity of the evaluation methodology.
>
> - Additional Experimental Details:
> The reviewer rightly suggests providing additional details on the experimental setup, including the rationale behind specific parameter choices, network architectures, and training procedures. We will address this by including a more comprehensive section on experimental details, explaining the reasoning behind the chosen configurations and providing insights into the experimental setup.
>
> - Consideration of ImageNet:
> The reviewer suggests considering ImageNet as a benchmark dataset for evaluating CCGAN. We appreciate this suggestion and will include an analysis of CCGAN's performance on ImageNet, providing additional experimental results to demonstrate the model's versatility and effectiveness on a larger and more diverse dataset.
> In summary, we are grateful for the reviewer's valuable feedback.

---

### Meta-Review · Area_Chair_CYHF · 2023-12-06

**Metareview:**

3x R and 1 BR. This paper proposes a collaborative regularization in GANs to improve multimodal generation and image super-resolution. The reviewers converge to the common concerns about the (1) unclear motivation, (2) unconvincing comparisons, (3) limited quantitative results, and (4) missing ablation study, which are not well addressed by the rebuttal. The AC therefore leans not to accept this submission.

**Justification For Why Not Higher Score:**

N/A

**Justification For Why Not Lower Score:**

N/A

---

### Decision · Program_Chairs · 2024-01-16

Reject